# Reward Models are Metrics in a Trench Coat

## Abstract

The emergence of reinforcement learning in post-training of large language models has sparked significant interest in reward models. Reward models assess the quality of sampled model outputs to generate training signals. This task is also performed by evaluation metrics that monitor the performance of an AI model. We find that the two research areas are mostly separate, leading to redundant terminology and repeated pitfalls. Common challenges include susceptibility to spurious correlations, impact on downstream reward hacking, methods to improve data quality, and approaches to meta-evaluation. Our position paper argues that a closer collaboration between the fields can help overcome these issues. To that end, we show how metrics outperform reward models on specific tasks and provide an extensive survey of the two areas. Grounded in this survey, we point to multiple research topics in which closer alignment can improve reward models and metrics in areas such as preference elicitation methods, avoidance of spurious correlations and reward hacking, and calibration-aware meta-evaluation.

## 1 Introduction

Reinforcement learning (RL) plays a major role in post-training, aligning, and adapting language models (LLMs) to a broad range of tasks (OpenAI, 2025; Comanici et al., 2025; xAI, 2025; Kimi et al., 2025; Guo et al., 2025). Scaling laws apply to reinforcement learning from human feedback (RLHF, Christiano et al., 2017) similarly as to the rest of the training stack (Bai et al., 2022a). As such, scalable alternatives to human feedback have become popular, either in the form of verifiable rewards (Lambert et al., 2024) or in the form of models that assess the quality of model outputs (Li et al., 2018). Developing robust and reliable *reward models* is crucial, as the downstream RL models can experience reward hacking (Amodei et al., 2016), optimizing for spurious correlations in the reward model rather than learning the intended behavior. To overcome these issues, reward models have experienced significant research interest.

In parallel to research on reward models, *model-based evaluation* of generated text has similarly seen a surge in interest, enabling a switch from "traditional" lexical metrics like BLEU (Papineni et al., 2002) and ROUGE (Lin, 2004) to learned models (Ma et al., 2018; 2019) and prompt-based approaches referred to as LLM-as-a-judge (Zheng et al., 2023). The two fields present two sides of the same coin: while reward models assess output quality to directly improve models, evaluation metrics assess output quality to identify potential areas of improvement. Both fields seek to develop classifiers that consume generated content as input and assign a goodness score as output. Both fields strongly benefit from rigor, consideration of the sociotechnical context in which a system is deployed, and improved correlation between model-based judgments and expert human raters. The key difference between the two is that while metrics tend to be more specialized, reward models tend to assess broad capabilities spanning many tasks. Due to their similarities, one might expect the fields learn from and inform each other, and that breakthroughs transfer quickly between them.

Our position paper argues that while this should be the case, it is not. Instead, the academic literature in these fields only infrequently informs each other and the fields are actively developing and using different terminology for the same methods. While metrics are commonly used to generate training data for reward models (Malik et al., 2025), and are thus instrumental to reward model performance, little attention is being paid to which metrics generate that data. We demonstrate this phenomenon by analyzing the citation graphs of papers in each sub-field, showing that inter-field citations account for fewer than 10% of total cited papers. We further support this claim by presenting results from two small experiments: one in which we apply a metric to a reward modeling benchmark and one

where we apply reward modeling techniques to a factuality evaluation benchmark. The results show that reward modeling approaches lag behind dedicated metrics for these specialized tasks, providing opportunities for improvements and motivating cross-testing on their respective benchmarks.

Motivated by these findings, we conduct an extensive survey of the two fields and their intersection. We lay out scenarios in which we can use all the tools at our disposal and showcase how it could lead to better reward models and evaluation metrics. Specifically, we argue that a closer collaboration could lead to major progress in overcoming reward hacking, in preference elicitation, and in meta-evaluation. We also discuss areas in which the fields differ and should not interact, and how this relates to Goodhart's law, which states that a measure ceases to be a good measure when it becomes a target. Grounded in these discussions, we make specific recommendations to researchers working on reward models and evaluation metrics on how the separation can be overcome.

## 2 How did modern Reward Models come about?

As part of the rising popularity of deep learning, RL started to be explored for tasks like structured prediction (Daumé et al., 2009; Ross et al., 2011), image recognition (Mnih et al., 2014; Ba et al., 2015) and for agents like the Neural Turing Machine (Zaremba & Sutskever, 2015). Successfully training a model via RL hinges on being able to generate reward signals. This includes being able to derive the value of intermediate states. As Sutton & Barto (2018) argue, "the most important component of almost all reinforcement learning algorithms we consider is a method for efficiently estimating values." Commenting on this issue, Yann LeCun famously criticized RL for having much sparser rewards than self-supervised learning during his talk "Predictive Learning" (LeCun, 2016).

This issue applies to generated language: generation has a combinatorially large state space with its sequential token choices from a large vocabulary, and no single objective number can represents the value of an output (Gehrmann et al., 2023). For that reason, generation models are typically trained via teacher-forcing, a supervised approach that shows the model a ground-truth token at each prediction step. This happens only during training, not at test-time. Moreover, while models are trained with a cross-entropy objective, they are evaluated via different metrics. Ranzato et al. (2016) coined the term *exposure bias* for this mismatch between training and test time.

If there was a way to directly optimize for the metric(s) we care about, the exposure bias could be overcome. Evaluation metrics are designed to act as a proxy for human judgments and are thus well-suited to serve as a reward function. While some inference-time methods optimize metrics (Wiseman & Rush, 2016; Freitag et al., 2021b), reinforcement learning is a natural fit to optimize for these metrics during training. REINFORCE (Williams, 1992) and minimum risk training (Duda & Hart, 1974) generate metric-based reward signals using sampled token sequences, and actor-critic approaches estimate partial rewards for predicted tokens (Bahdanau et al., 2017). Various instantiations of these approaches were used for machine translation (Ranzato et al., 2016; Shen et al., 2016), image captioning (Rennie et al., 2017), video captioning (Pasunuru & Bansal, 2017), and summarization (Paulus et al., 2018).

At that time, the reward models were measuring lexical overlap between a generated sequence and a ground truth (e.g., Papineni et al., 2002; Lin, 2004; Vedantam et al., 2015). These metrics have well-understood drawbacks (e.g., Reiter, 2018; Freitag et al., 2020), especially for RL (Choshen et al., 2020). Among others, they lead to *reward hacking* where models generate non-fluent language that maximizes reward scores (Amodei et al., 2016). Researchers worked to overcome these issues, for example by regularizing the training process by combining cross-entropy losses with RL or by hand-crafting additional reward functions (Pasunuru & Bansal, 2018; Kryściński et al., 2018; Wu et al., 2018a). The advent of metrics measuring semantic rather than lexical similarity led to significantly reduced reward-hacking since the new models avoided over-optimizing for the generation of relevant words without fluent context (Li et al., 2018; Yasui et al., 2019; Scialom et al., 2019).[1] These models led to a clear path whereby new metrics could be validated and then used as reward models. For example, the metric BLEURT was introduced (Sellam et al., 2020), evaluated as part of the WMT Metrics shared task (Mathur et al., 2020), and then assessed as reward model (Shu et al., 2021).

---

[1]This is also related to Generative Adversarial Networks (Goodfellow et al., 2014) for generation (e.g., Yu et al., 2017; Wu et al., 2018b) where a discriminator differentiates generated text from the ground truth, thus similarly generating a model-based signal for human-likeness.

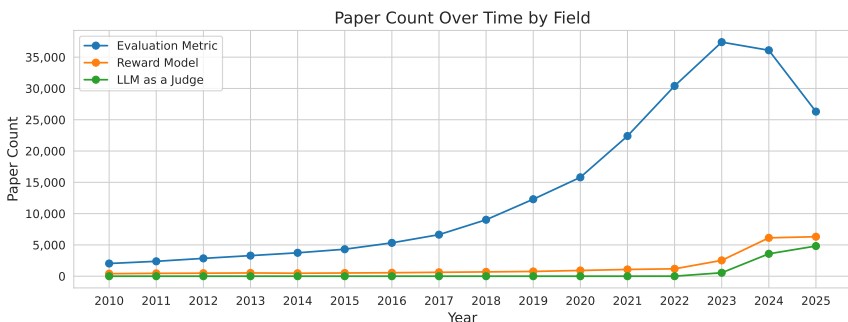

Figure 1: The figure shows the number of publications per year in the three subfields according to a keyword search on Google Scholar. Publications on evaluation metrics have slowed, even though research on reward modeling and LLM-as-a-judge is quickly rising in popularity. If the fields were actively learning from one another, one could assume that mentions of "evaluation metrics" should be growing alongside these newly emerging fields, but they are not.

In parallel to the work above, the concept of *reinforcement learning from human feedback (RLHF)* was introduced for game playing and robotics (Christiano et al., 2017). In an essay titled "Scalable agent alignment via reward modeling: a research direction", Leike et al. (2018) propose capturing human preferences via dedicated *reward models*. This research culminated in the work on RLHF for summarization (Stiennon et al., 2020) which popularized Proximal Policy Optimization (PPO, Schulman et al., 2017) as RL approach for text generation. While Stiennon et al. (2020) analyzed correlations between ROUGE and their human preference data, they did not use widely accepted alignment metrics, existing human preference corpora, or the semantic similarity evaluation metrics discussed above. In the followup work on InstructGPT (Ouyang et al., 2022), there are no references to the generation RL literature and no evaluation of the reward model. Subsequent work introduced the notion of AI Feedback as reward models (Bai et al., 2022b; Lee et al., 2024a) and argued that language model probabilities can be directly used to model rewards in a Bradley-Terry model (Rafailov et al., 2023; Bradley & Terry, 1952). Neither draws the connection to the role that perplexity and model probabilities played in existing evaluations (e.g., Lewis et al., 2020; Min et al., 2023).

This separation culminated in benchmarks for reward models (e.g., Frick et al., 2025; Lambert et al., 2025; Liu et al., 2025c; Zhou et al., 2025) and for metrics (e.g., Honovich et al., 2022; Clark et al., 2023; Freitag et al., 2024) that exist in parallel without meaningful interaction. This raises the question of whether this disconnect is part of a broader trend. And if the two fields were integrated tighter, would we be in a better state today? And what should one learn from the other?

# 3 QUANTIFYING THE RESEARCH FIELD SEPARATION

Figure 1 establishes the need for this investigation by showing the number of papers found on Google Scholar per year that contain the exact strings "Evaluation Metric", "Reward Model", and "LLM-as-a-judge". We include "LLM-as-a-judge" as an emerging field that has similarly experienced rapid growth and which also uses language models to estimate the quality of generated output. Notably, despite the exponential growth of the two emerging topics, the number of papers mentioning evaluation metrics decreased in 2024, with the trend continuing into 2025.

If the terminology was merely changing, one would expect the new literature to still build on the older one. For that reason, we empirically study the cause of this phenomenon by conducting a citation analysis. We select up to 300 papers per field per year (2021–2025) via the Semantic Scholar Graph API, with sensitivity checks at 100/200 yielding similar trends (Kinney et al., 2023).[2] For each paper, we additionally retrieve its citations, yielding approximately 10,000 citations per year for each field to analyze. As a proxy to identify whether a paper in field A cites a paper in field B, we

---

[2]Documented at https://api.semanticscholar.org/api-docs/graph. The search results in the maximum 300 papers for the first two fields and 8, 25, and 43 papers respectively for LLM-as-a-judge over the past three years. More details on this analysis in Appendix A.

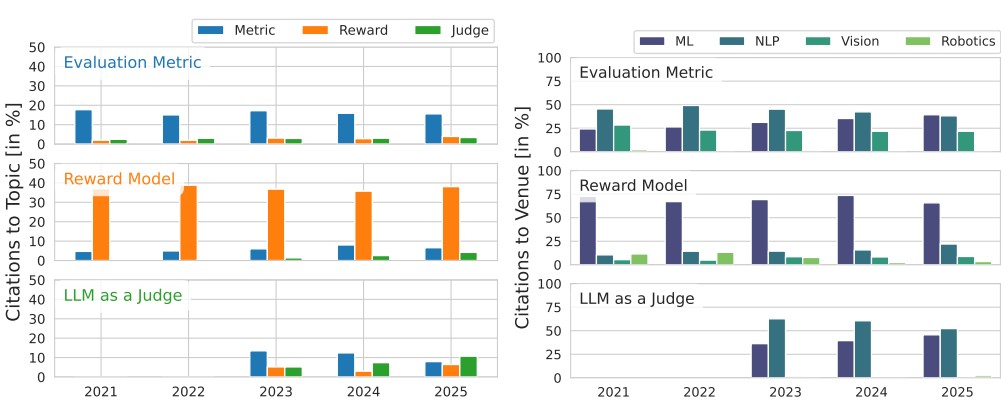

(a) The fraction of citations from one field to another, based on keywords in cited papers.

(b) The fraction of citations to papers in venues associated with a research area.

Figure 2: In our analysis of citation dynamics across the three fields, we find that evaluation papers tend to cite other evaluation papers across research fields, while reward model papers mostly cite each other and are highly focused on machine learning venues. LLM-as-a-judge work mostly cites ML and NLP venues, but has less clear citation dynamics.

select *signaling terms* for each field: (1) "metric(s)", (2) "reward", "reinforcement", "policy", and (3) "judge". If any of those terms appears in the title or abstract of a cited paper, we count this as an inter-field or intra-field citation. The results in Figure 2a show that evaluation metrics and reward models are distinct fields, with only few inter-field citations but many intra-field citations. This is especially pronounced for reward models where almost 40% intra-field citations. The numbers for evaluation metrics trend lower at around 15–20% which we attribute to the heterogeneity of the field; for example, papers on metrics for summarization cite summarization papers rather than only evaluation papers. LLM-as-a-judge is an outlier, with too few papers to draw definitive conclusions.

We find more evidence for the field separation when we analyze the venues of the cited papers. For this, we categorize venues into fields (e.g., ICLR as ML venue) and calculate the percentage of citations to papers published in the various fields. The results in Figure 2b reveal that reward model research predominantly cites research in machine learning venues and not NLP and Computer Vision. In contrast, evaluation metric work is evenly distributed and LLM-as-a-judge work focuses on ML and NLP.[3] Since all observed trends are stable across years, we conclude that the three research fields are largely separate with limited interaction.

## 4 WHAT CAN ONE LEARN FROM THE OTHER?

A rebuttal to our proposition that the two fields should learn from each other is that maybe there is little to learn. We thus highlight two scenarios in which a closer cross-field interaction could have changed conclusions or yielded additional insights.

### 4.1 METRICS CAN PERFORM WELL ON REWARD MODEL BENCHMARKS

The recently introduced RewardBench-M (Gureja et al., 2025) uses a subset of the MAPLE dataset (Zhu et al., 2024) to assess reward models on translation evaluation. The task requires identifying which of two translation outputs was rated higher by human evaluators. The data is split into an easy and difficult subset based on the difference of human scores of the two provided translations. While all their tested models perform nearly perfectly on the easy subset, Gureja et al. (2025) remark that "*models that perform well on easy tasks can struggle to maintain the same level of performance on harder translations, indicating the need for more sophisticated mechanisms to handle [...] challenging scenarios*". However, no machine translation evaluation metric was assessed as a

---

[3]We omit fields with $\leq 5\%$ of citations in all years, including Speech, IR, and HCI. Ambiguous venues like preprint servers or broad venues like AAAI are excluded from this analysis. Detailed list in Appendix C.

|  | de→en | en→de | zh→en | en→zh |
|---|---|---|---|---|
| GPT-4o | **71.0** | 61.0 | **77.0** | 80.0 |
| Aya Expanse 32B | 62.0 | **69.0** | 76.0 | 79.0 |
| CometKiwi-DA (2022) | 59.0 | **68.0** | 59.0 | **86.0** |

Table 1: Results on the hard machine translation evaluation subset of RewardBench-M. For non-English evaluations, a 3 year old model with 550M parameters outperforms much larger LLMs.

baseline. Thus, to test this hypothesis, we evaluate the three-year-old metric CometKiwi (Rei et al., 2022) which is based on InfoXLM (Chi et al., 2021) and has only 550M parameters.

The results of CometKiwi alongside the two best-performing models on the challenging translation test set of RewardBench-M (Dang et al., 2024; Hurst et al., 2024) are shown in Table 1. Despite its age and being significantly smaller, CometKiwi performs similarly on German and outperforms the other models on Chinese, with the overall best evaluation performance for the non-English generated text, demonstrating that the "sophisticated mechanisms" needed in current reward models already exist. Building on this observation, the MetaMetrics approach (Anugraha et al., 2024) has been evaluated on the latest MT metrics shared task (Freitag et al., 2024) and on the RewardBench leaderboard (Lambert et al., 2025), scoring highly in both, although not with the same model.

## 4.2 Reward Models can underperform on Metrics Benchmarks

Another area in which reward model and metrics benchmarks are aligned in their goals is the assessment of how well models can assess factuality and attribution. There exist benchmarks for metrics (Honovich et al., 2022), model performance (Jacovi et al., 2025), and reward models (Malik et al., 2025) that assess factuality. Recent work demonstrates the effectiveness of LLM judge (and reward) models (e.g., Calderon et al., 2025; Hashemi et al., 2024), some even finding that dedicated finetuned evaluation models underperform LLM judges (Huang et al., 2025).

Among these benchmarks, the metrics benchmark SEAHORSE (Clark et al., 2023) is the largest with over 100,000 human judgments of summarization quality aspects across multiple languages. For this experiment, we prompt various LLMs with the same instructions provided to human annotators in SEAHORSE to give a binary judgment whether a summary is attributable to an article.[4] Due to a lack of data availability, we exclude the WikiLingua (Ladhak et al., 2020) subset of SEAHORSE and focus only on XLSum (Hasan et al., 2021) and MLSum (Scialom et al., 2020), retaining 7,793 of the 18,330 test examples. We report Pearson correlation ($\rho$) and accuracy.

The results (Table 2) show that LLMs underperform the dedicated model trained on in-domain data. This remains true even if we assess judge models with a high reasoning budget like Gemini 2.5 Pro and GPT-5. In fact, the two reasoning models have an 89% agreement rate, higher than the inter-rater agreement of 73% reported in the paper, indicating that the models look for similar input and output features to make their prediction. The results are fairly consistent across languages. Interestingly, all models score lowest on English among the evaluated languages.

Overall, our findings disagree with Huang et al. (2025) who show that LLM judges can outperform dedicated metrics in similar setups, while agreeing with Bavaresco et al. (2025) who show low LLM judge correlations for summarization evaluation. Multiple explanations exist for the results presented here, including annotation artifacts that cause a lower performance of the LLM judge setup. However, we can conclude that for evaluating attribution for summarization, it remains unclear whether LLM judges have caught up to dedicated models, a question that requires further rigorous study. This conclusion mirrors the argument by Chehbouni et al. (2025) that the "rapid and widespread adoption [of LLM judges] may have occurred prematurely". Moreover, as shown in Table 2, the LLM judge setup outperforms a strong Natural Language Inference model (NLI) baseline (Conneau et al., 2018). As such, this setup could still be useful in cases where dedicated training data for a reward model or evaluation metric is unavailable.

---

[4]We optimized performance on the validation set to minimize the effect of the prompt format.

|  | $\rho$ | Acc. % |
|---|---|---|
| ROUGE-L | 0.13 | |
| mT5$_{XNLI}$ | 0.43 | |
| mT5$_{SEAHORSE}$ | 0.59 | |
| GPT-4o | 0.47 | 73.3 |
| Gemini 2.0 Flash | 0.42 | 72.1 |
| Gemini 2.5 Flash | 0.48 | 73.8 |
| Claude Sonnet 4 | 0.45 | 70.1 |
| *+ Reasoning* | | |
| Gemini 2.5 Pro | 0.50 | 73.9 |
| GPT-5 | 0.47 | 70.2 |

Table 2: Pearson $\rho$ coefficient and binary prediction accuracy on SEAHORSE for identifying whether a summary is **attributable** to a source article. The baselines are finetuned mT5$_{XXL}$ models by Clark et al. (2023). The LLM-as-a-judge approach is outperformed by a dedicated trained metric.

## 5 METRICS AND REWARD MODELS ARE (NOT) THE SAME

Metrics and reward models both judge quality aspects of generated content with the goal of being aligned with human preferences. Yet, they are not the same: they can differ in their design, application, training, and testing. To explore these aspects, we provide a survey of the two fields and discuss themes where a closer interaction could lead to mutually helpful insights.

### 5.1 DESIGNING REWARD MODELS AND EVALUATION METRICS

**Sociotechnical Context matters** Evaluation metrics tend to be narrowly focused on specific quality aspects. These quality aspects should follow clear and standardized definitions such that the metrics are transferable and produce scores that are understandable across organizations. A lack of transparency in how metrics are designed and the subsequent lack of reproducibility has been subject of much past criticism (Rankel et al., 2013; Post, 2018; Gehrmann et al., 2023).

In contrast, if a reward model is the provider of training signals during reinforcement learning, it must therefore be able to score a myriad of tasks and output types. As such, modeling human preferences encompasses many aspects of preference beyond output quality, including whether a model correctly refuses undesired requests or avoids producing toxic language (Bai et al., 2022b). These judgments depend on the specific application the model is used for and the policies governing this application. Reward models that measure these application-specific aspects are inherently less transferrable and tied to the specific organizations that develop them (Gehrmann et al., 2025). Following this reasoning, these reward models are inherently not comparable to another, which calls into question the utility of non-specific reward-modeling benchmarks.

**Aspect-aligned reward models** Fine-grained assessments of (partial) generations are areas with extensive recent work in RL that more closely align with work on evaluation metrics (Gunjal et al., 2025; Lightman et al., 2024). It is of particular interest, since a diverse set of reward signals can mitigate issues that arise from single-objective optimization (Freitag et al., 2021b; Zhang et al., 2024; Fisch et al., 2024). A popular approach for this is to use reward models that score rubrics instead of providing generic preferences (Gunjal et al., 2025). Rubrics are fine-grained evaluation criteria (Arora et al., 2025; Hashemi et al., 2024), similar to those traditionally assessed by dedicated metrics. Rubric-based prompted scoring, alongside learned reward models, is mentioned as an instrumental ingredient for post-training of models like Gemini 2.5 (Comanici et al., 2025).

Successfully assessing rubrics requires clear definitions of the evaluation categories (Howcroft et al., 2020). Yet, even for popular concepts like "hallucination rate", definitions can vary widely (Maynez et al., 2020; Rashkin et al., 2023; Ji et al., 2023). Increasing the consistency of these definitions will be crucial as reward models become more specific, and thus are designed more similar to evaluation

metrics where the topic of fine-grained assessments is well-studied (e.g., Eyal et al., 2019; Wang et al., 2020; Fabbri et al., 2021; Scialom et al., 2021; Lee et al., 2024b; Wei et al., 2025).

## 5.2 TRAINING REWARD MODELS AND EVALUATION METRICS

**Data Collection**  The data collection methodology for any model must be aligned with its design goals. For reward models, this means reflecting the preferences of the intended audience of the downstream model, which can be extremely broad. Since many aspects of generated text cannot be objectively assessed, this necessitates collecting feedback from diverse sources (Casper et al., 2023; Metz et al., 2025). The culture and lived experience of raters can lead to drastically different subjective preference judgments (e.g., Aroyo et al., 2023; Rastogi et al., 2024; 2025).

Another critical question to consider is whether the selected raters have sufficient expertise, as changes in annotation quality can lead to drastically different insights (Freitag et al., 2021a; Wei et al., 2024). In many cases, existing metrics and reward models already outperform non-expert raters, and only the highest quality annotations can further improve the models (Cui et al., 2023; Liu et al., 2024; Wen et al., 2025b). Moreover, Wen et al. (2025a) find that RL may produce errors that are increasingly difficult for humans to detect. However, hiring raters with expertise to judge long-form generation is notoriously challenging (Zhang et al., 2023).

**Optimization targets**  Design aspects such as access to a ground truth and the output format (pairwise preferences, categorical labels, or continuous scores) influence how models are developed. These choices depend on the downstream use case, and can have significant impact on model efficacy. For example, reference-less evaluation has improved significantly in recent years but still underperforms reference-based metrics (Ma et al., 2019; Freitag et al., 2024). Similar results were found for LLM-as-a-judge setups (Krumdick et al., 2025). These choices are reflected in benchmarking practices; many reward models produce pairwise comparisons, and their benchmarks consequently focus on this binary setup (Frick et al., 2025). In contrast, metrics typically generate continuous outputs, allowing for more flexible evaluation. By focusing primarily on pairwise judgments, reward model development may be ignoring the potential benefits of continuous scoring.

Another shared goal is the development of lightweight models that can run efficiently alongside larger models during inference or training. Advances in distillation, quantization, parallelization, and pruning are therefore highly relevant to both fields. Consequently, approaches to model compression that seek to train student models to outperform their teachers can equally benefit the development of both reward models and evaluation metrics (Kim et al., 2024; Sun et al., 2023).

## 5.3 TESTING REWARD MODELS AND EVALUATION METRICS

**Identifying and Debugging Reward Hacking**  A lack of correlation between reported reward model and downstream RL model performance has been attributed to limitations of the reward model (Ivison et al., 2024; Kim et al., 2025; Wen et al., 2025c). When the reward model does not robustly generalize, or focuses on spurious correlations, it can lead to *reward hacking*. Amodei et al. (2016) describe reward hacking as the process of "gaming" flaws in the reward model to maximize the rewards without learning the intended behavior. This phenomenon was empirically observed for text (Pasunuru & Bansal, 2017; Kryściński et al., 2018; Wu et al., 2018a) and non-text RL (Amodei & Clark, 2016; Krakovna et al., 2020; Nagarajan et al., 2021).

It is not specific to reward models, as most classification models suffer from spurious correlations (Ribeiro et al., 2016; McCoy et al., 2019) and spurious correlations were found in reward models (Liu et al., 2025b) and metrics (Sun et al., 2019). Among the effects, reward models may prefer more confident-sounding answers (Leng et al., 2025), exhibit a verbosity bias (Saito et al., 2023), focus more on style than content (Feuer et al., 2025), and results may be confounded by the order in which outputs are shown (Wang et al., 2024). Relatedly, the problem of *sycophancy* has been characterized as models learning to match user beliefs over generating truthful responses (Sharma et al., 2024). Murugadoss et al. (2025) and Hu et al. (2024) further show that the detail of LLM-as-a-judge prompts have little influence on its performance, implying that models rely too much on their implicitly learned quality criteria definitions. These issues motivate work on diagnostic datasets (Gabriel et al., 2021), distractor generation (Qiu et al., 2020; Dhole et al., 2023), and model interpretability (Jacovi et al., 2023), to become aware of and overcome spurious correlations.

**Meta-Evaluation Frameworks** The field of meta-evaluation is concerned with the question of how we evaluate evaluators. Callison-Burch et al. (2007) popularized this practice in NLP through a shared task series that performs a yearly assessment of MT metrics. Meta-evaluation measures two aspects: *segment-level* and *system-level* performance. A high system-level performance means that system rankings in a leaderboard are trustworthy, while segment-level assessments look at whether individual pairs of system outputs are ranked correctly. These two measures are not always correlated (Wei & Jia, 2021), motivating an approach that matches how a model is used.

Algorithms like DPO (Rafailov et al., 2023) use the reward score difference between a chosen and rejected model output as training signal. This directly matches the segment-level meta-evaluation. However, a known issue is that evaluation metrics are often not well-calibrated (Kocmi et al., 2024), which may cause issues if they are applied as reward models. Moreover, reward model benchmarks like RewardBench 2 (Lambert et al., 2025; Gureja et al., 2025) do not consider score calibration, instead reporting overall accuracy on the task of identifying the highest rated system output, which more closely matches a system-level assessment. As a result, calibration issues may be overlooked if one focuses only on reward model benchmark performance. This oversight of segment-level assessments could further contribute to the lack of correlation between reward model and downstream model performance. Thus, future work on reward model benchmarking could benefit from reporting segment-level rather than system-level performance, including assessments of score calibration.

**Meta-Evaluation Targets** A complicating factor for the meta-evaluation of reward models is the breadth of tasks for which they need to assess output quality. Their meta-evaluations thus need to strike a balance of breadth, validity, and relevance. Ivison et al. (2024) suggest that existing reward model benchmarks are too narrow, especially considering their performance variance across tasks (Bavaresco et al., 2025). Benchmarks like RewardBench 2 already average multiple categories, but the question of how to aggregate sub-category scores into a single ranking becomes important. To that end, Frick et al. (2025) find that pessimistic reward model evaluations instead of average performance are more indicative of downstream model performance, motivating alternative leaderboard designs that focus on finding shortcomings, rather than averaging performance numbers. These issues are further exacerbated when the systems that are being evaluated by evaluation metrics and reward models improve. As these models become harder to distinguish, biases in the evaluation setup become more noticeable (Wei & Jia, 2021) and tie handling procedures need to be introduced (Thompson et al., 2024; Sun et al., 2025).

## 5.4 RECOMMENDATIONS

While developing an unhackable reward model is likely impossible (Skalse et al., 2022), metrics, and more directly reward models, share a symbiotic relationship with the downstream models where improvements in one translate to improvements in the other (Gehrmann et al., 2021). This means, we should strive to produce the most accurate estimate of human preferences. To that end, both fields benefit from having high-quality training and meta-evaluation data. This data needs to be grounded in clear definitions in the sociotechnical context that the to-be-assessed models are deployed in. While it is unavoidable to introduce spurious correlations, in both fields it is crucial to identify and measure them and to mitigate their impact on downstream uses.

Modeling choices and optimization targets similarly align between the two fields, whether that is applying LLM-as-a-judge, or training classification models on human-curated data. Due to this overlap, newly introduced methods for modeling human preferences should be evaluated on metrics and reward model benchmarks alike to paint a more accurate and complete picture of how these methods perform. More generally, meta-evaluation and the development of leaderboards is an area in which the fields have significantly diverged. They should come together to address the poor correlation between reward model benchmark scores and downstream model performance. Shared best practices on tie handling, segment-level correlation measures, conducting model calibration assessments, and collecting test datasets will benefit both fields.

However, as reward modeling matures as a field, it will be important to avoid falling into traps like developing default models and benchmarks that, despite being outdated, continue to be broadly used (Bommasani & Cardie, 2020). Instead, evaluation research could adopt the practice from reward modeling of moving to new and better performing model as they become available.

While reward models and evaluation metrics should be developed using the same best practices, one cannot be used to replace the other. As Goodhart's law states, *when a measure becomes a target, it ceases to be a good measure* (Goodhart, 1984). Applying to both fields, models may perform well as a proxy for the distribution over human preferences but will diverge in the tail (i.e., rarely seen model inputs) and may over-generalize and focus on spurious patterns (Manheim & Garrabrant, 2018; Gao et al., 2023). Similar arguments apply to the utility of shared tasks and leaderboards which similarly are a frequent target of criticism (e.g., Scott & Moore, 2006; Ethayarajh & Jurafsky, 2020; Thomas & Uminsky, 2020; Raji et al., 2021; Bowman & Dahl, 2021). Our practical recommendation is thus for the fields to share insights into methodologies, but not to collapse into one field.

A full collapse would risk creating a monoculture where only a few benchmarks dictate model optimization, rather than having many different targets (Koch & Peterson, 2024). As Singh et al. (2025) state, an "over-reliance on a single leaderboard creates a risk that providers may overfit to the aspects of leaderboard performance, without genuinely advancing the technology in meaningful ways". Issues with broadly adopted evaluation setups lead to overspecialization and lack of generalization beyond what a specific leaderboard measures (Liu et al., 2025a; Zouhar et al., 2024). Being too rigid in how meta-evaluations are conducted could also exclude new methods from being investigated fairly (Perrella et al., 2024).

Furthermore, while methods for reward modeling may be informed by insights from evaluation metric development, the specific reward models may not perform well on the same benchmarks. As discussed above, reward models are often specific to a sociotechnical context, and would thus not perform well on public reward modeling benchmarks. This may cause a rift between industry and academic research where the best reward modeling approaches are not publicly disclosed because they are too entangled within this context. Yet, especially for models that measure human preferences for whether a model output is considered offensive or undesirable, it is critical to develop public standards and be transparent about the underlying policies a model is trying to enact.

## 6 CONCLUSIONS

In this work, we have argued that evaluation metrics and reward models share many similarities. Their developers need to make the same choices about their inputs and outputs, their collection of training and validation data, and the resulting models suffer from the same drawbacks. While the application areas and the specific choices made during development may differ, at their core, both seek to model human preferences of model output quality. This supports our thesis that the fields should look at and learn from each other's advances, rather than continuing to exist in parallel.

We grounded this discussion in a citation analysis that demonstrated that the research fields are developing mostly in isolation from each other. This separation of fields can lead to missed opportunities, the rediscovery of established findings, and potentially flawed conclusions. We quantified the separation through two experiments that show that reward models may be lacking when assessed on domains for which evaluation metrics are already available. We provided an extensive survey and discussed several areas in which future work on metrics and reward models, meta-evaluations, and benchmark creation could incorporate insights from both fields. While we recommend against the development of a monoculture with too few relevant benchmarks, we encourage researchers to consider work from both fields and work on unifying both methodologies and terminologies.

Beyond the scope of this work, we acknowledge efforts in reinforcement learning to solve tasks with verifiable rewards, for example math problems (Ke et al., 2025), for which reward models play a less central role. In this domain, model-based reward modeling approaches largely perform string matching between the verified and generated answer, and thus do not require as complex approaches as those discussed here. Training models for these verifiable domains can induce reasoning capabilities and has led to broader generalization (Guo et al., 2025; Comanici et al., 2025). While this finding does not make reward models for non-verifiable domains obsolete, it presents a possible alternative or parallel path in which reward models do not play such a central role. Moreover, we note that improvements in reward models may not always translate into downstream model improvements. The *superficial alignment hypothesis* by Zhou et al. (2023) poses that the reinforcement learning stage primarily changes 'how' a model responds, rather than contributing new world knowledge. Thus, even a perfect reward model cannot overcome fundamental knowledge gaps from pre-training.

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

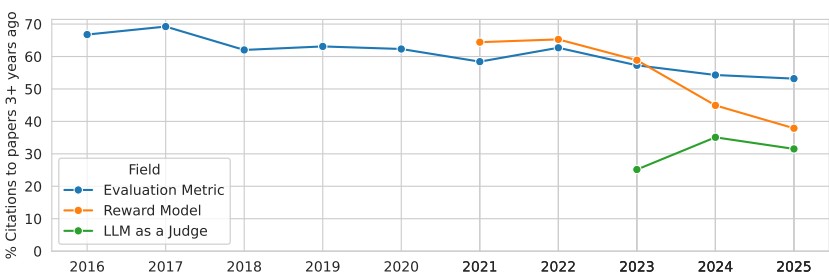

Figure 3: We show the percentage of citations to papers that were published more than three years ago. Reward model literature exhibits outlier behavior in which this percentage is decreasing drastically every year.

## A   DETAILED METHODS FOR CITATION ANALYSIS

The results in the main text present a high-level overview of the key findings of our citation analysis. We chose to present results for the last 5 years since reward models that resemble those discussed in this paper were only popularized in this time frame. While the Semantic Scholar API yields results for the years prior, they are mostly irrelevant to the discussion at hand. We extended the analysis for evaluation metrics back to 2016 but find no significant difference in results. Therefore, we omit them for readability.

Similarly, we choose to present results of an analysis of up to 300 papers per year as a result of qualitative assessment of the data. A qualitative assessment found that, beyond the first 300 papers, search results became too noisy, with irrelevant papers being retrieved. We repeated the analysis with only the top 100 and 200 papers with no significant differences in results.

The specific keywords for the analysis in Figure 2a were selected based on repeated trials to maximize precision at the cost of recall. For example, including "evaluation" as a proxy keyword for evaluation metrics would have yielded any paper that discusses their evaluation results, not necessarily discusses how to build evaluation metrics. Similarly, we included generic RL-related terms like "policy" for reward models since the terminology was evolving and papers only fairly recently converged on this term. To not miss citations to relevant papers prior to 2020, we included them at the risk of overestimating the true citation count.

As a result, the specific numerical results are a side effect of this keyword-based identification and should be interpreted with caution. While an LLM-based identification process may yield more accurate results, it would require processing a significant number of tokens. Since we were only interested in aggregate trend information, we found the results from keyword-based searches sufficient and stable across many variants.

## B   RECENCY BIAS IN CITATIONS

We quantify recency bias in citations across the three fields of study. Citations only to recent papers would provide an additional piece of evidence that insights from work before LLMs became popular are not being considered. Indeed, we find that while the average age of a cited paper for evaluation metrics published in 2025 is 5.0 years, cited papers by reward modeling and LLM-as-a-judge papers are only 3.6 and 3.8 years old. 62.2% of citations in reward modeling papers are to papers published less than 2 years ago (68.5% for LLM-as-a-Judge), in contrast to 46.8% for evaluation metrics. Critically, Figure 3 shows how citations to older papers have been decreasing, especially in literature on reward models. This result is an indicator that reward modeling research is evolving quickly and that benchmarks are quickly made irrelevant by new results.

## C    ASSIGNMENTS OF CONFERENCE TO SUBFIELD

For our analysis of citations to research areas in Section 3, we assign academic venues to an area if the venue is clearly affiliated with it. For example, AAAI's scope is all of AI and we therefore do not include it in this analysis. We include a venue in this analysis if papers published there received at least 50 citations among all the 50,000+ papers included in our analysis. We account for various misspellings, capitalization differences, and abbreviations, but only list each venue once in the following list of assignments:

**Machine Learning**    COLT, ICLR, ICML, JMLR, NeurIPS / NIPS, TMLR, TNNLS

**Natural Language Processing**    ACL (including Findings of ACL), CONLL, EACL, EAMT, EMNLP, INLG, LREC, NAACL, SemEval, WMT

**Robotics**    CoRL, ICRA, IROS, IJRR, RSS

**Vision**    CVPR, ECCV, ICCV, IJCV, MICCAI, TIP, TOG, WACV

## D    USE OF LLMS FOR WRITING

After manually writing the entire draft, we used ChatGPT and Google Gemini to improve its writing. Specifically, we prompted them to find spelling and grammatical errors and suggest improvement to sentence structures. Additionally, we prompted them to provide feedback on whether the argument structure was clear throughout all paragraphs, asking for specific improvements can can be made. All suggestions were manually reviewed.

