# OpenReview forum: "Reward Models are Metrics in a Trench Coat"
_ICLR.cc/2026/Conference — Submitted to ICLR 2026_

### Official Review · Reviewer_TGxA · 2025-10-30

**Soundness:** 3
**Presentation:** 4
**Contribution:** 3
**Rating:** 6
**Confidence:** 4

**Summary:**

The paper discusses reward models and evaluation metrics, advocating that the communities developing these respective methods could learn more from each other. This position is supported by (1) citation analysis showing the lack of citations between the two communities, (2) empirical analysis showing how existing evaluation metrics can outperform reward models on benchmarks used to evaluate reward models, highlighting how prior work has been ignored.

**Strengths:**

* The paper is well written, and provides a nice survey of related work. This could be useful, especially for newer folks in these fields.
* While I had thought that the similarities between evaluation metrics and reward models were common knowledge, the paper's citation analysis and empirical analysis support their key claims that these respective communities could benefit from greater knowledge sharing.

**Weaknesses:**

* Given that this is primarily a position paper, the value of the contribution is rather subjective. But I thought the main claims were well argued, and there is likely an audience at ICLR that would find this paper interesting.
* Some differences, while covered to some degree in section 5, could be expanded on with respect to the different usages of reward models and evaluation metrics, and therefore the different demands on their accuracy and robustness. In particular, reward models operate in a potentially more adversarial setting with respect to reward hacking, and therefore demands on robustness are potentially greater. Additionally, an evaluation metric may be useful if it provides sufficient correlation with human judgements aggregated over an entire eval. However, as reward models need to provide directional signal for individual prompts, the accuracy demands are potentially higher. Might be nice to have some discussion of this and if this could cause divergence in methods.
* Super nit: inconsistent capitalization/casing of paragraph headers.

**Questions:**

None

---

> ### Author Response · Authors · 2025-11-24
>
> Thank you for your review of this paper, and your view that there is an audience at ICLR that would find the paper interesting. Expanding on some of the differences between the two types of models is a great idea to use the additional space for a camera-ready version of this paper and we had to omit some of these points for space reasons.
>
> Your suggestion is actually extremely relevant for the distinction between system-level and segment-level correlation measures for evaluation metrics. As you state, in many cases we may only care about averaged (or system-level) indications of performance of a model, but in many other cases we may care more about individual examples on which a model performs well.
> The latter is much harder to perform well on, and we can expand on how work on segment-level evaluation could potentially inform better reward model development.

---

### Official Review · Reviewer_izC9 · 2025-11-01

**Soundness:** 2
**Presentation:** 2
**Contribution:** 1
**Rating:** 2
**Confidence:** 4

**Summary:**

The paper is a position paper that attempts to tie together two concepts - reward models and model evaluation (LLM-as-a-judge). Authors point out that these two fields rarely talk and could benefit from learning from each other.

Authors support their ideas with a cross-citation study and some cross-over experiments

**Strengths:**

1. The paper ties together two important concepts (a) reward models and (b) model-based evaluation using something like LLM-as-a-judge
2. The paper does a good job of representing recent work in both fields.

**Weaknesses:**

1. I have a problem with RLHF being mentioned as a reinforcement learning method. RLHF is not really RL, since learning doesnt happen in a dynamic environment with feedback
2. In the 1st paragraph of the paper, the authors mention that scaling laws for RLHF exist. However, very few convincing works exist in this direction
3. The motivation for the paper in the 1st paragraph is quite weak. RLHF requires good reward models, so reward modeling is necessary. This could have been the simple motivation to start the paper, no need to talk about scaling laws.
4. Table 1 is missing variance / error bars, making results untrustworthy. Machine translation evaluation is supposed to be notoriously difficult to pin down without repeat runs.
5. I dont get what I can really take away from this paper. If the experiments in Tables 1 and 2 were a lot more rigorous and comprehensive, then the claims in the paper would have some solid data. However, the current set of experiments are insufficient.

**Questions:**

How reliable is Figure 1? How did the authors create it? Were similar terms taken into consideration?
Can you consider running a broader set of experiments for cross-over?

---

> ### Author Response · Authors · 2025-11-24
>
> Thank you for sharing your concerns about our first paragraph and first figure. We address them in order below:
>
>
> 1) RLHF is not really referring to any particular reinforcement learning method here, but rather the concept of learning from human feedback. As you point out, many actual implementations use some form of offline RL or data filtering instead of dynamic environments. We don’t think this distinction is relevant for this paper.
> 2) We will add more references to papers that mention that scaling RL during post-training is critical for broader generalization of model performance. We can also remove the mention of scaling laws in particular since this point is not relevant to our paper
> 3) As per our answer to (2), we are happy to simplify the argument.
> 4) As per our answer to reviewer YigS, we are improving the topic part of our citation analysis. We are unclear what error bars would represent in this case or what the uncertainty would be over
> 5) We refer to our discussion and conclusion for concrete takeaways. Our experiments are illustrating the points made throughout for additional anecdotal evidence. It is unclear to us what specifically you think they are insufficient for.
>
> As mentioned in our responses to reviewers YigS and UcTw, we will strengthen the setup for our argument in the introduction and add an additional classifier-based analysis for the topic-wise citations.

---

### Official Review · Reviewer_UcTw · 2025-11-02

**Soundness:** 2
**Presentation:** 2
**Contribution:** 3
**Rating:** 4
**Confidence:** 4

**Summary:**

This position paper argues that reward models (RMs)—particularly those used in RLHF and often implemented as LLM-as-a-Judge—are essentially evaluation metrics in disguise. The author supports this claim through a citation analysis showing limited cross-pollination between the RM and metric communities, and presents two empirical experiments suggesting that specialized metrics can outperform general-purpose RMs on narrow tasks (e.g., machine translation quality or summarization factuality).

**Strengths:**

- The paper raises an important and timely observation: the conceptual and methodological overlap between reward modeling and automatic evaluation is underappreciated.
- The citation analysis is compelling and highlights a real fragmentation in the literature.
- The call for shared benchmarks, meta-evaluation practices, and terminology alignment is well-motivated and could benefit both communities.

**Weaknesses:**

see questions

**Questions:**

While I largely agree with the paper’s core observation—that RMs and metrics share the same goal of approximating human judgment—I believe the paper underestimates a fundamental architectural and functional distinction between the two.

Traditional evaluation metrics (e.g., BLEURT, COMET, QAFactEval) are domain- or task-specialized: they are designed, trained, and validated for a specific generation task (e.g., MT, summarization) and often rely on reference texts, fine-grained rubrics, or narrow definitions of quality (e.g., faithfulness, fluency). This specialization allows them to achieve high correlation with human judgments *within their scope*—as the paper correctly demonstrates.

In contrast, reward models are inherently general-purpose by design. From their inception in RLHF (e.g., Stiennon et al., 2020; Ouyang et al., 2022), RMs were meant to operate across a vast, open-ended space of user intents, domains, and safety constraints—from coding and math to dialogue, instruction-following, and refusal behavior. They must balance helpfulness, harmlessness, truthfulness, style, verbosity, and more, often without reference outputs.

This generality is not a bug but a feature required by the RLHF pipeline. One cannot simply replace a scalar RM with a collection of specialized metrics in practice, because:
1. Engineering complexity: Integrating dozens of heterogeneous, reference-dependent, domain-specific metrics into a unified reward signal for policy gradient updates is nontrivial and brittle.
2. Differential vulnerability to hacking: Each metric has its own failure modes (e.g., BLEU favors repetition, NLI-based metrics are sensitive to paraphrasing). An RL agent could exploit inconsistencies across metrics or over-optimize one at the expense of others.
3. Scalability bottleneck: The very promise of RLHF is to scale alignment via automated preference signals. Reverting to a patchwork of handcrafted metrics undermines this vision and impedes end-to-end training at scale.

Thus, while I agree that specialized metrics can—and should—inform RM design (e.g., via rubric-based scoring, calibration-aware training, or diagnostic datasets), I disagree with the implication that RMs are “just metrics” that can be swapped out for existing ones in real-world alignment pipelines. Rather, reward models represent a *generalization* of the metric paradigm: they are *systemic, reference-free, multi-dimensional, and deployment-aware evaluators*—a necessary evolution for aligning general-purpose LLMs.

---

> ### Author Response · Authors · 2025-11-24
>
> Thank you for your review and for sharing your concerns about the premise of our paper. We want to clarify that we share your concerns and it was not our intention to suggest that we should replace reward models with metrics or even a mixture of metrics.
>
> However, we slightly disagree with points 2 and 3:
>
> - Point 2 is about reward hacking. While it is true that metrics are susceptible to reward hacking, there has been significant work invested into making metrics more robust and providing transparency into their failure modes. We have not seen the same level or rigor applied to reward models. Our paper provides an opportunity for researchers working on reward models to identify opportunities which of these methods may be worth applying themselves.
> - Point 3 is about scalability. The argument here is that we shouldn’t revert to a patchwork of metrics rather than a single reward model since it impedes end-to-end training. We think our argumentation in the paper may have led to a misunderstanding. Rather than saying that we should use a patchwork of metrics instead of the current approaches, we argue that there are opportunities to **improve the end-to-end approaches**, for example, by incorporating some more targeted metrics training data, by identifying shortcomings, and by improving how reward models are being benchmarked.
>
> As such, we fully agree with the statement of the final paragraph that reward models are a generalization of the metric paradigm. This is a very nice framing we will happily incorporate into the structure of our arguments.
> As we point out in the paper, there is a gap between the two research fields where insights are not being generalized to the degree that they could be.

---

> > ### Comment · Reviewer_UcTw · 2025-11-26
> >
> > > there has been significant work invested into making metrics more robust and providing transparency into their failure modes.
> >
> > Could u plz elaborate this?
> > I think currently the most important part of reward model is reward hacking, but i am not familiar with metric hacking ( It is something like use some repeat phrase to hack BLEU?)

---

> > > ### Author Response · Authors · 2025-12-03
> > >
> > > Hi UcTw - happy to elaborate! Since you don’t have the ability to respond anymore, I will try to be exhaustive here.
> > >
> > > “Metric Hacking” as you refer to it is basically a model exploiting weaknesses in a metric that make the output seem better than it actually is. “Traditional” hand-crafted metrics like BLEU/ROUGE are particularly susceptible to this phenomenon since they have no ability to measure semantics and instead focus on lexical similarity. But the same happens to modern metrics in exactly the same way that reward models can be exploited.
> > >
> > > Take for example the analysis by Gabriel et al. (2021) (https://aclanthology.org/2021.findings-acl.42) who performed a meta-analysis of whether metrics can capture various forms of factual mistakes. They find that standard evaluation metrics for summarization are not valid measures of factuality.
> > > Maynez et al. (2020) (https://aclanthology.org/2020.acl-main.173/) ran a comprehensive analysis of what models / metrics accurately identify hallucinations in question answering and summarization, equally finding that both traditional and more modern metrics fail at the task.
> > > Grounded in these previous analyses, Honovich et al. (2022) (https://aclanthology.org/2022.naacl-main.287) worked toward solving these issues by developing entailment models that perform better at measuring factual consistency.
> > > This line of work also led to the SEAHORSE dataset by Clark et al. (2023) we analyze in our paper (https://aclanthology.org/2023.emnlp-main.584) and which shows further improvements over these entailment approaches by collecting relevant in-domain data for attribution.
> > > Gehrmann et al. (2023) (https://jair.org/index.php/jair/article/view/13715/26927) provide a survey of additional ways to probe for and analyze flaws in metrics in Section 3.4.
> > >
> > > In parallel to these studies, there was also significant work on developing standardized definitions of what exactly a metric should capture. The above references work by Maynez et al. introduced the concepts of intrinsic and extrinsic hallucinations to differentiate between information that is incorrect given the input to a model and information that may be incorrect because it is not grounded in context at all. Focusing on the former, Rashkin et al. (2023) develop the concept of “attribution to identified sources” (https://aclanthology.org/2023.cl-4.2/) which allows for the development of metrics that are more clearly defined. SEAHORSE uses that definition as well.
> > >
> > > More recent work blurs the lines a lot more, as LLM-as-a-judge has been applied for both evaluation and as reward modeling technique. However, a lot of the recent papers we cite in our submitted paper analyze LLM-based judges with the goal to make them more robust and faithful to some definition of model output quality.

---

### Official Review · Reviewer_YigS · 2025-11-05

**Soundness:** 2
**Presentation:** 3
**Contribution:** 3
**Rating:** 6
**Confidence:** 4

**Summary:**

The paper raises an interesting discussion on the reward models in reinforcement learning with human feedback. Given that the reward models are the human preference proxies for diverse domains, such as good summaries and safe conversations, the paper revisits systematic metrics that are not modeled as neural models. Interestingly, applying such metrics to human preference benchmarks shows that those metrics are not in a distinct field, in fact, they are on par with reward models.

**Strengths:**

1. The paper revisits the overlooked overlap between evaluation metrics and reward models, addressing the RLHF community’s tendency toward self-referential benchmarking, which is a timely perspective in the era of RLHF and post-training.
2. The extensive survey across both reward modeling and evaluation metrics provides a valuable resource for researchers and well supports the context of the paper.
3. The illustrative experiments effectively demonstrate the claim that specialized metrics can outperform large reward models on targeted tasks.

**Weaknesses:**

While the paper makes a compelling conceptual argument, several aspects could be strengthened for more rigor and completeness:

- **Reward models as human proxies for RLHF training**: The argument could be more convincing if it examined how the proposed equivalence between metrics and reward models propagates to downstream RLHF outcomes. Reward models’ true purpose is to serve as accessible proxies for human preferences during RL training, not merely as static evaluators. Demonstrating whether a metric-based proxy actually induces comparable policy behavior (e.g., in translation or summarization) would make the thesis empirically more robust, in addition to analyses like Table 1.
- **Credibility of the citation analysis**: Figure 2 is presented as the central quantitative justification for Section 3, claiming the field separation between evaluation metrics, reward models, and LLM-as-a-Judge. However, it is questionable if the keyword filtering is sufficient for capturing the conceptual overlaps between the fields. For instance, papers on LLM-as-a-Judge often use “metrics” as a keyword for how they measure performance, not as an evaluation metric [1]. Given that this figure grounds the central claim of disciplinary isolation, such methodological looseness could overstate the degree of separation between the two fields.

&nbsp;

**References**

[1] Zheng et al., 2023, “Judging LLM-as-a-Judge with MT-Bench and Chatbot Arena.” (NeurIPS 2023)

**Questions:**

- Reward models and evaluation metrics may both approximate human preference functions, but they differ in their use within training loops. What behavioral differences would you expect if we used each as the training signal for identical RLHF objectives? Either logical analysis or empirical demonstrations would be helpful.

---

> ### Author Response · Authors · 2025-11-24
>
> Thank you for the suggestion about demonstrating that metrics, when used as a reward model, can lead to improved results. This was actually shown as part of the WMT shared task 2025 for which the overview paper (https://aclanthology.org/2025.wmt-1.22/) was just released. The application of metrics here is actually two-fold: (1) multiple approaches use metrics during an RL step, for example as data filter for DPO (see https://aclanthology.org/2025.wmt-1.55.pdf). (2) Many approaches use metrics during **decoding** via minimum bayes risk (see https://arxiv.org/pdf/2411.03524). Together, these methods lead to some of the most powerful models for machine translation. We will add a discussion of this to the paper.
>
> We further agree that part of our citation analysis is somewhat limited due to reliance on keywords. To mitigate this, we ran a lot of versions of keyword-based searches and added the more definite venue-based analysis. While we cannot change the Semantic Scholar API itself, we will make the topic-analysis of citations more robust by adding an additional prompt-based classifier that runs on titles and abstracts and identifies whether a cited paper belongs to one of the three categories we consider.
>
> To address your question how metrics, when used as reward models, would change downstream model behaviors: As mentioned above - for tasks for which these metrics were designed, it would lead to better quantitative performance. However, we don’t want to argue that metrics should blindly be used in such a manner if the target is to develop a general-purpose LLM. In those cases, we argue that the way we develop and evaluate reward models should take lessons from the research field of evaluation metrics since this field already tackled many of the challenging questions.

---

### Meta-Review · Area_Chair_Kvu3 · 2026-01-07

**Summary:**

This paper argues that reward models and evaluation metrics are functionally equivalent but developed by separate research communities, leading to redundant work and repeated pitfalls.

After the rebuttal, the reviewers still raise several concerns, for example, the experiments are insufficient, limited in small domains. Also, the paper underestimates fundamental differences between task-specific metrics and general-purpose reward models. RMs handle diverse intents, multiple quality dimensions. The experiments/statement in Section 4 only narrowly on machine translation tasks. As a position paper, the work does not provide sufficiently concrete or insightful takeaways for the community. The conceptual overlap between metrics and reward models, while interesting, is not developed into clear guidance or novel perspectives that would advance either field.

**Reviewer Concerns:**

see metareview

**Reviewer Scores:**

see metareview

---

### Decision · Program_Chairs · 2026-01-26

Reject